# Flaw Classification Algorithm for Heat Exchanger Tubes Using a Bobbin-Type Magnetic Camera

**Sunbo Sim [1], Hoyong Lee [2], Heejong Lee [1] and Jinyi Lee [1,*]**

[1] IT-based Real-Time NDT Center, Chosun University, Gwangju 61452, Korea; ssb3842@gmail.com (S.S.); happybell221@naver.com (H.L.)

[2] Department of Defense Science and Technology, Gwangju University, Gwangju 61743, Korea; hylee@gwangju.ac.kr

[*] Correspondence: jinyilee@chosun.ac.kr; Tel.: +82-62-230-7101

**Abstract:** This paper presents an algorithm that estimates the presence, location, shape, and depth of flaws using a bobbin-type magnetic camera consisting of bobbin probes and a bobbin-type integrated giant magnetoresistance (GMR) sensor array (BIGiS). The presence of the flaws is determined by the lobe path of the Lissajous curves obtained from bobbin coil with respect to the applied frequency. The location of the flaw, i.e., whether it is an inner diameter (ID) or outer diameter (OD) flaw, can be determined from the rotational direction of the lobe with respect to the frequency change. The shape of the flaw is then determined from the area of the lobe and the BIGiS image. At this stage, multi-site damage can be determined from the BIGiS image. The effectiveness of the flaw classification algorithm was evaluated using various types of artificial flaws introduced into small-bore tube test specimens made of austenitic stainless steel.

**Keywords:** flaw classification algorithm; bobbin-type integrated GMR sensor array (BIGiS); time-varying electromagnetic field (T-EMF); heat exchanger tube

---

## 1. Introduction

Nuclear power plants consist of many heat ex-changers (HXs), such as a steam generator (SG), moisture separator and reheater (MSR), condenser (CDS), and feedwater heat exchanger (FWH). Among them, the high pressure (HP) FWH is made of welded austenitic stainless steel tubes (ASTM A688-TP304L). A unit of HP FWH contains three heaters and consists of more than 13,000 tubes. The tubes have small bores and significant length, with an outer diameter (OD) and thickness of 15.87 mm (0.625 in) and 1.27 mm (0.05 in), respectively. Few nondestructive testing (NDT) methods exist which can quickly assess the integrity of this material and its structure, namely paramagnetic, narrow, long, and large number of tube bundles.

Eddy current testing (ECT) can be applied to small-bore tubes, does not require a couplant between the sensor and the specimen, and can inspect a large number of HX tubes within a short time. Additionally, it has the advantage of the detection of various types of flaws easily, including volumetric flaws [1]. However, it poses a challenge when trying to distinguish crack-like flaws from volumetric flaws, and multisite damage is difficult to evaluate.

Furthermore, if the NDT results show the presence of any crack in the tube, the tube must be plugged regardless of the repair criteria; however, tubes with volumetric flaws should be plugged only if its thickness exceeds the criteria. Therefore, recognizing the shape of the flaw and quantifying its depth is important so that a crack is not mistaken for a volumetric flaw.

Lee et al. developed bobbin-type magnetic cameras to quickly access the large number of HX tubes [2–8]. In order for magnetic camera technology to be effectively applied to industrial field,

an algorithm should be provided to make it easier for inspectors to identify and evaluate flaws. This study is a follow-up project of the bobbin-type magnetic camera technology development by Lee et al. It proposes an algorithm to identify the location, recognize the shape, and quantitatively evaluate the depth of the flaws when inspecting HP FWH tubes of austenitic stainless steel using a bobbin-type magnetic camera. The effectiveness of the proposed algorithm was evaluated using three test specimens that represent different types of flaws, ASME test standard, and multisite damage.

## 2. Principles

### 2.1. Detecting the Presence and Location of Flaws Using Differential-Type Bobbin Coils

A differential-type bobbin coil measures the signal difference between two adjacent coils. The time-varying electromagnetic fields (T-EMFs) generated by applying an alternating current (AC) to the bobbin coils induces a current in a conductive tube. If the tube has no flaw, the T-EMFs (facing each other) will cancel each other out so that the output of the differential signal, i.e., the impedance and phase angle, is zero. On the other hand, if there is a flaw in the tube, the intensity and phase angle of the induced current are affected as each coil passes over the flaw. As a result, the impedance difference between the first and second coil is reversed in sign as the coils pass through the flaw, and the phase angle indicates a 180° difference. The phase angle is important in determining the location of the flaw because it exhibits different tendencies depending on whether the flaw is located at the inner diameter (ID) or outer diameter (OD). In addition, the impedance and phase angle change with the depth of the flaw, and can be used to quantitatively evaluate the depth of the flaw.

### 2.2. Measuring Alterating Magnetic Fields Using GMR Sensors

When a primary T-EMF is applied to the bobbin coil as described above, an induced current and a secondary T-EMF are generated in the conductive HX tube. By the same principle, when the primary T-EMF in the opposite direction is applied to adjacent differential-type bobbin coils, the secondary T-EMF is canceled in the center of the coils and this generates an annular region where the output becomes zero. This secondary T-EMF balance collapses as it passes over the flawed area, as shown in Figure 1. At this instance, if a device that measure the magnetic flux density (MFD), such as a GMR sensor, is placed in the center of the differential-type bobbin coils and scanned inside the tube in the axial direction (z-direction), the MFD distribution can be measured.

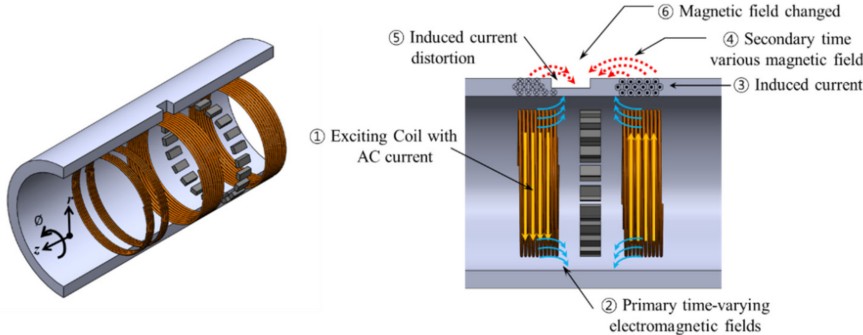

**Figure 1.** 3D illustration of coil and sensor array of bobbin-type magnetic camera and magnetic flux density measurement principle around a flaw.

The electrical signal measured by a GMR sensor is represented by Equation (1) [9]. $V_G$ represents the output voltage of a GMR sensor. The orientation of the GMR sensor is sensitive to changes in the z-axis MFD, resulting in the measurement of $B_Z$, which is the MFD in the z-direction. $C_1$ and $C_2$ are constants representing the intrinsic characteristics of the GMR sensor, and $p$ is the input voltage. Equation (1) shows that the output of the GMR sensor increases nonlinearly with respect to the MFD regardless of the magnetic poles.

$$V_G(r, \phi, z) = C_1 \times p \times (1 - e^{-B_z^2}) + C_2 \tag{1}$$

Furthermore, because of the flaw in the tube, the T-EMF changes both the amplitude phase angle of the output signal. The amplitude and phase angle can be calculated by multiplying the output signal obtained by Equation (1) with the input signal (sin $\omega$) and the 90° leading input signal (cos $\omega$), respectively, and then integrating the results. Figure 2 shows the procedure for calculating the real and imaginary parts of the output signal by synthesizing the cosine and sine signals. The real part, imaginary part, amplitude and phase angle of the output signal can be calculated using Equations (2)–(5). In this study, the amplitude and phase angle difference distributions measured by the GMR sensors along the inner wall of the tube specimen were graphed on a two-dimensional plane with the sensor array axis ($\Phi$) and longitudinal (scan) axis ($z$).

$$Re(r, \phi, z) = \int_0^{2\pi} V_G(r, \phi, z) \sin \omega t \, dt \tag{2}$$

$$Im(r, \phi, z) = \int_0^{2\pi} V_G(r, \phi, z) \cos \omega t \, dt \tag{3}$$

$$Z(r, \phi, z) = \sqrt{Re(r, \phi, z)^2 + Im(r, \phi, z)^2} \tag{4}$$

$$\Phi(r, \phi, z) = tan^{-1} \frac{Im(r, \phi, z)}{Re(r, \phi, z)} \tag{5}$$

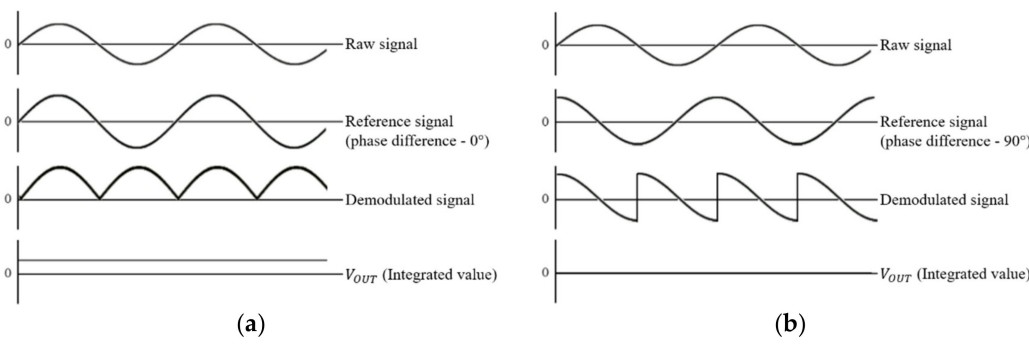

(**a**)　　　　　　　　　　　　　　　　　　　　　　　　　　(**b**)

**Figure 2.** Real and imaginary part of the output signal calculation procedure: (**a**) real part; (**b**) imaginary part.

## 3. Experiment and Discussion

### 3.1. Test Specimens

Figure 3 shows the test specimens with artificial flaws used to verify the effectiveness of the proposed inspection system. The specimens were made of stainless steel (SS304) with an ID of 13.3 mm and a thickness of 1.27 mm. The first specimen had different types of artificial flaws: tapered wear (A), flat wear (B), flat bottom hole (C), axial notch (D), circumferential notch (E), dent (F), and steam cut (G). The second specimen had through holes (H, I), OD notch pairs (J, K, L, M), and ID notch pairs (N, O, P). The OD and ID notch pairs which represent multisite damage were arranged with axial notches and circumferential notches of different depths facing each other in the circumferential direction.

The third specimen was an American Society of Mechanical Engineers (ASME) standard test specimen, in which flat bottom holes (Q, R, S, T, U) and grooves (V, W) were introduced. Table 1 shows the flaw shape, position, and depth of each test specimen.

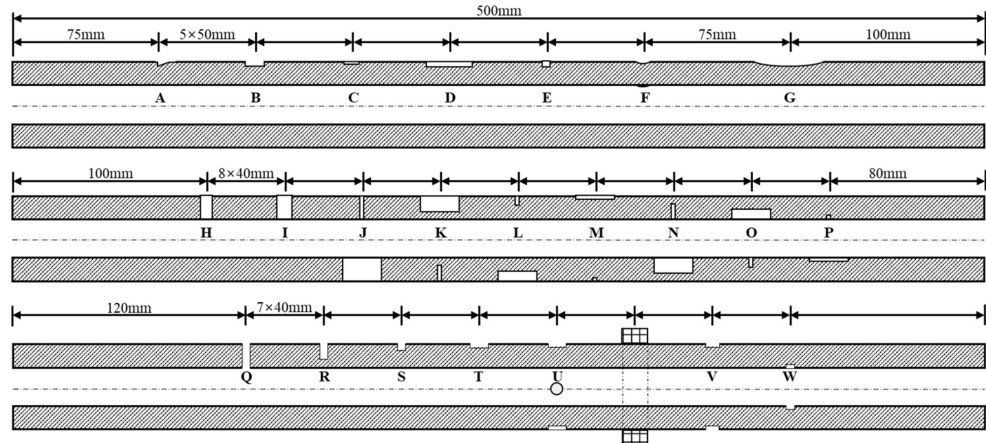

**Figure 3.** Test specimens (unit: mm).

**Table 1.** Specifications of sensors.

| Location | Shape | Width/Length [mm] | Depth [mm] | Depth [%] |
|---|---|---|---|---|
| A | 180° TW [1] | 6.32 | 0.508 | 40 |
| B | 180° FW [2] | 6.32 | 0.508 | 40 |
| C | FBH [3] | 2.778 | 0.254 | 20 |
| D | OD AN [4] | 0.127/9.0 | 0.508 | 40 |
| E | OD CN [5] | 0.127/9.0 | 0.508 | 40 |
| F | Dent | 5/10 | 0.127 | 10 |
| G | SC [6] | 10/40 | 0.254 | 20 |
| H | FBH | 1.0 | TWH [7] | 100 |
| I | FBH | 1.76 | TWH | 100 |
| J | OD CN/OD AN | 0.127/9.0 | TWH | 100 |
| K | OD AN/OD CN | 0.127/9.0 | 0.762 | 60 |
| L | OD CN/OD AN | 0.127/9.0 | 0.508 | 40 |
| M | OD AN/OD CN | 0.127/9.0 | 0.254 | 20 |
| N | ID CN/ID AN | 0.127/9.0 | 0.762 | 60 |
| O | ID CN/ID AN | 0.127/9.0 | 0.508 | 40 |
| P | ID CN/ID AN | 0.127/9.0 | 0.254 | 20 |
| Q | FBH | 1.321 | TWH | 100 |
| R | FBH | 1.984 | 1.016 | 80 |
| S | FBH | 2.778 | 0.762 | 60 |
| T | FBH | 4.763 | 0.508 | 40 |
| U | FBH | 4.763 | 0.254 | 20 |
| V | OD groove | 3.175 | 0.254 | 20 |
| W | ID groove | 1.588 | 0.127 | 10 |

[1] Tapered wear; [2] flat wear; [3] flat bottom hole; [4] axial notch; [5] circumferential notch; [6] steam cut; [7] through wall hole.

## 3.2. Experimental Setup

Figure 4 shows the experimental setup and the 3-D illustration of the sensor probe. Differential-type bobbin coils were placed in front of and behind the sensor probe. The bobbin coils located at the front (hereinafter referred to as front bobbin coils) had 192 turns and were 1.5 mm wide, and the gap between the two coils was 2 mm. The bobbin coils located in the rear (hereinafter referred to as rear bobbin coils) had 180 turns and were 3 mm wide, and the gap between the two coils was 8 mm. In the center of rear bobbin coils, 22 GMR sensors were arranged in a circumferential direction. The distance between the sensors was 1.5 mm with and angular resolution of 16.36 degrees.

The front bobbin coils were activated with a constant-voltage driven AC of 50, 100, and 150 kHz, and the rear coils were activated with a constant-voltage driven AC of 30 kHz. The output (amplitude and phase angle) of the front bobbin coils and GMR sensors was obtained through a 300 kHz low pass filter, an AC amplifying circuit (16 dB for the front bobbin coils, 53.89 dB for the GMR sensors), and a real and imaginary component circuit. A sensor probe was inserted into the specimens and scanned using an automated pusher-puller of 50 mm/sec axial speed.

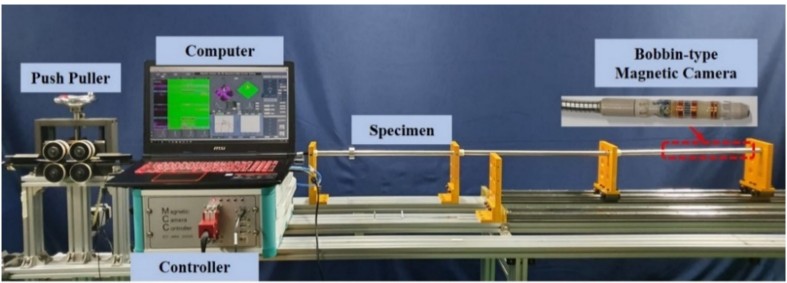

**Figure 4.** Experimental setup.

### 3.3. Test Results

Figures 5–7 show the output signals of the front bobbin coils according to the flaw types (A–G). For all flaws that changed in volume, the Lissajous curves rotated clockwise as the input frequency increased. However, for the dent, which did not change in volume, the curve rotated counterclockwise. Volumetric flaws also showed the lobe path of the Lissajous curve starting at the fourth quadrant and ending at the second quadrant as the front bobbin coils passed over the flaws, as shown in Figure 8. However, the dent had a lobe path starting at the second quadrant and ending at the fourth quadrant. The alphabets in the top right of the figures represent the flaw locations listed in Table 1.

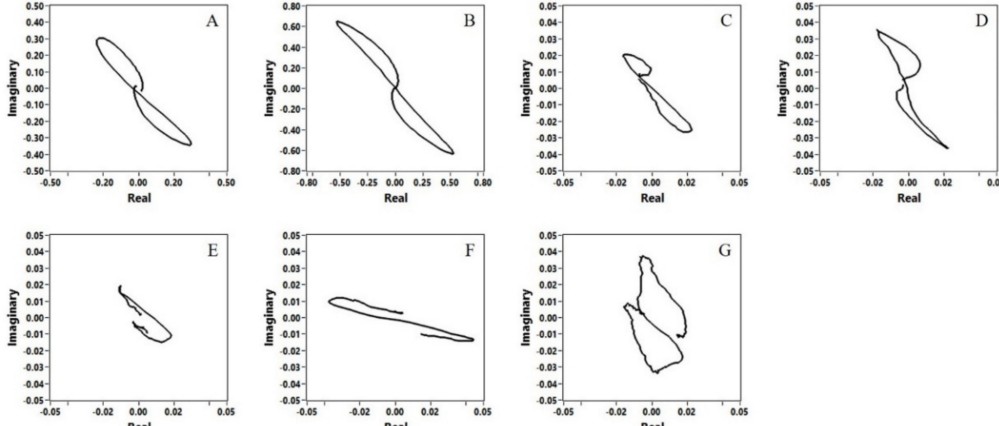

**Figure 5.** Lissajous curves from the front bobbin coils for the specimen with different type flaws (50 kHz AC).

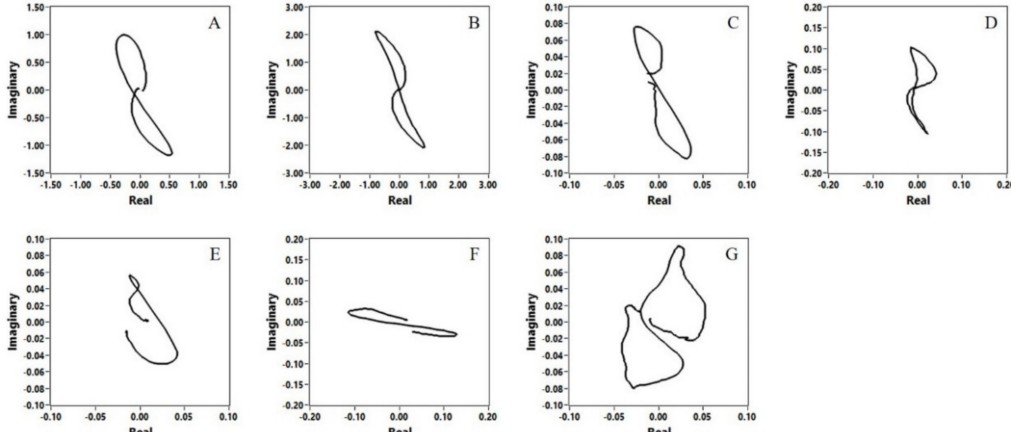

**Figure 6.** Lissajous curves from the front bobbin coils for the specimen with different type flaws (100 kHz AC).

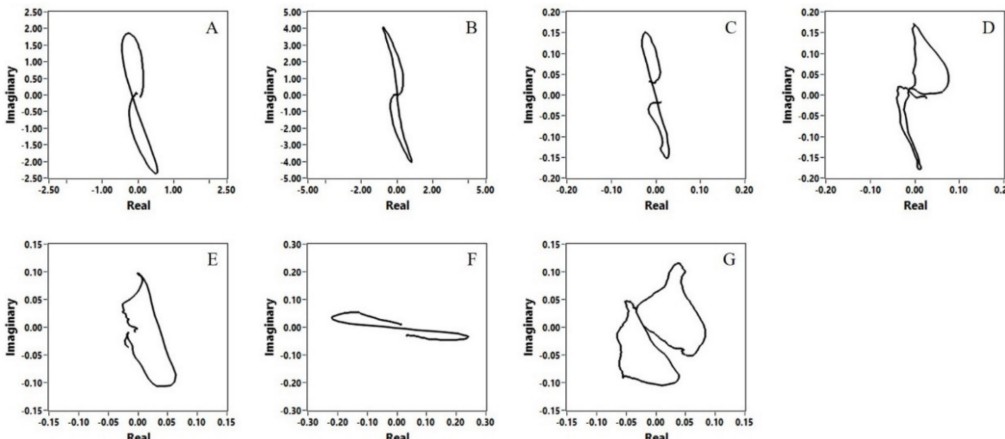

**Figure 7.** Lissajous curves from the front bobbin coils for the specimen with different type flaws (150 kHz AC).

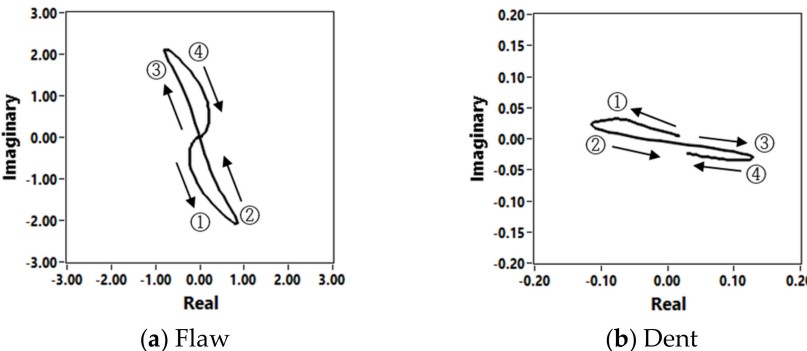

(**a**) Flaw　　　　　　　　　　　　　　(**b**) Dent

**Figure 8.** Comparison of lobe path direction.

Figure 9 shows the 2D images of the T-EMF distribution around the artificial defects (A–G) obtained through the GMR sensor array. Flaws with large-area distribution on the tube wall, such as tapered wear, flat wear and steam cut, were also widely distributed in the T-EMF. In the case of OD axial notch, the T-EMF distribution was long in the axial direction. The T-EMF distribution of the OD circumferential notch was similar to that of a flat bottom hole (FBH), but the lobe area of the Lissajous curve through the front bobbin coils output was smaller than those of other flaws, and thus the flaw shape could be inferred accordingly.

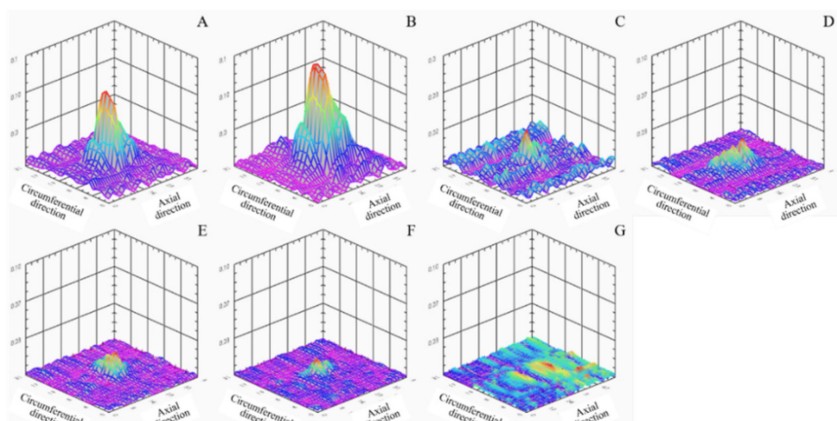

**Figure 9.** 3D image of the T-EMF from the BIGiS for the specimen with different type flaws (30 kHz AC).

Figures 10–12 show the output signals of the front bobbin coils for the artificial defects on the second specimen (FBH and multisite damages). For through-wall holes (TWHs) and 100% notches,

the lobe did not rotate at higher frequencies. However, the lobe can be seen to have been rotated clockwise by the OD flaws and counterclockwise by the ID flaws. Additionally, all flaws (TWH and OD or ID notch pairs) produced the lobe path of the Lissajous curve shown in Figure 9. However, the shapes of the lobes obtained by the front bobbin coils were not much different from the single damages in Figures 5–7, even though they were multisite damages. Therefore, the multisite damage could not be known from the output of the front bobbin coils. However, more than 40% OD notch pairs and more than 20% ID notch pairs were clearly identified through the T-EMF images obtained through the BIGiS as shown in Figure 13.

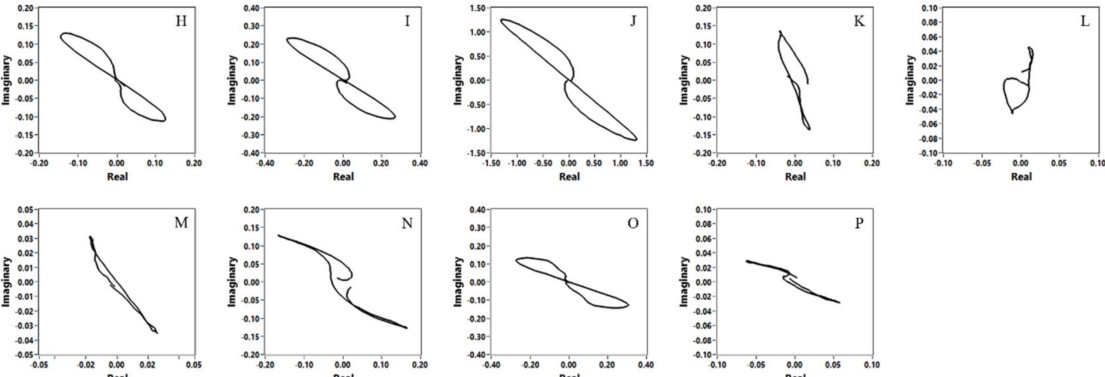

**Figure 10.** Lissajous curves from the front bobbin coils for the specimen with multisite flaws (50 kHz AC).

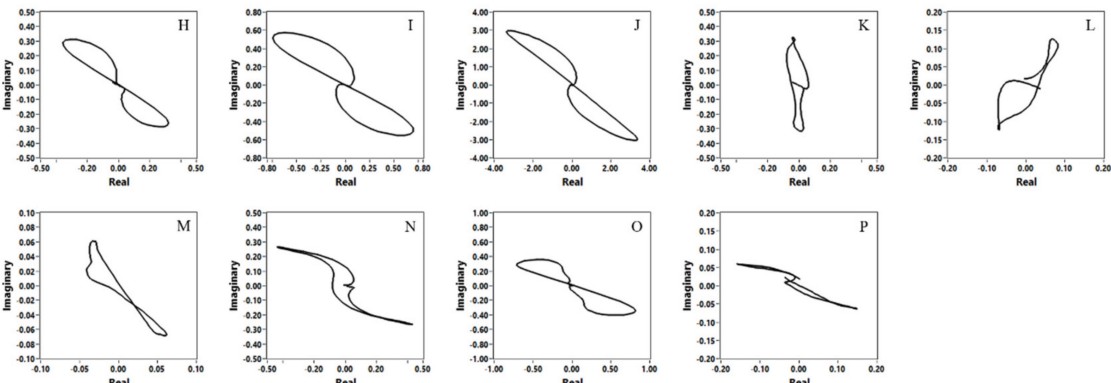

**Figure 11.** Lissajous curves from the front bobbin coils for the specimen with multisite flaws (100 kHz AC).

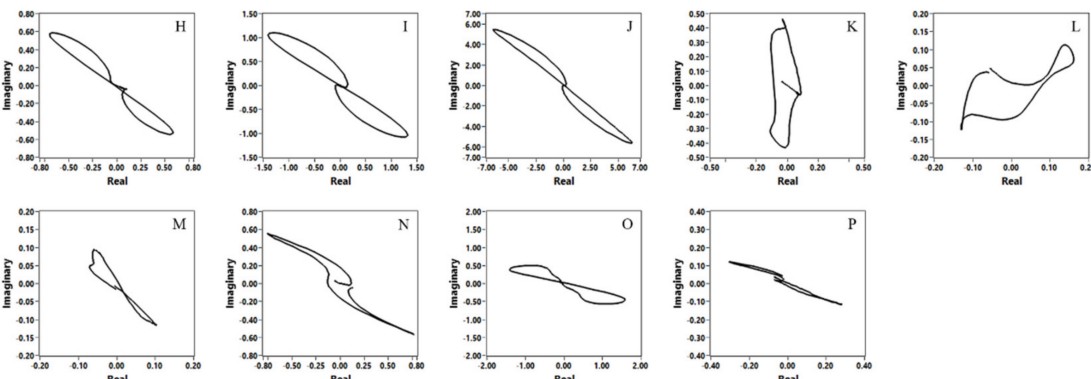

**Figure 12.** Lissajous curves from the front bobbin coils for the specimen with multisite flaws (150 kHz AC).

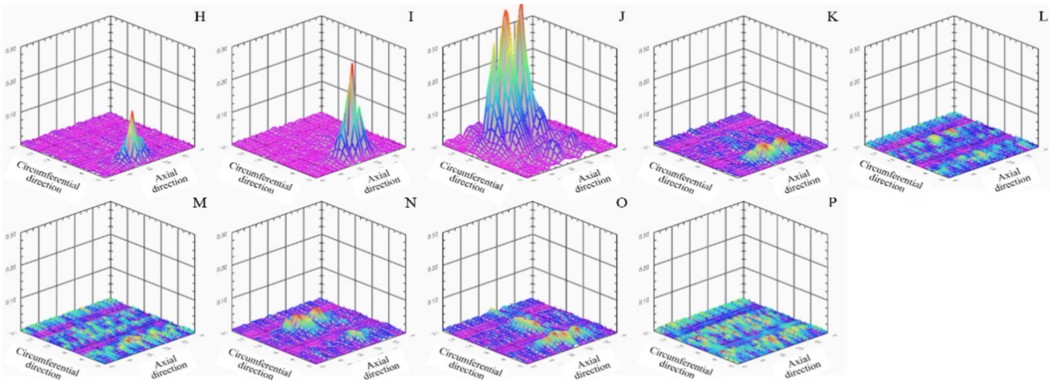

**Figure 13.** 3D image of the T-EMF from the BIGiS for the specimen with multisite flaws (30 kHz AC).

Figures 14–16 show the output signals of the front bobbin coils for the artificial flaws in the ASME standard specimen. The lobes of the Lissajous curves rotated clockwise as the input frequency increased for OD flaws, and counterclockwise for ID flaws. In the same manner, all flaws produced the lobe paths shown in Figure 9. As shown in Figure 17, the 2D images of the T-EMF distribution identified all flaws in the ASME standard specimen.

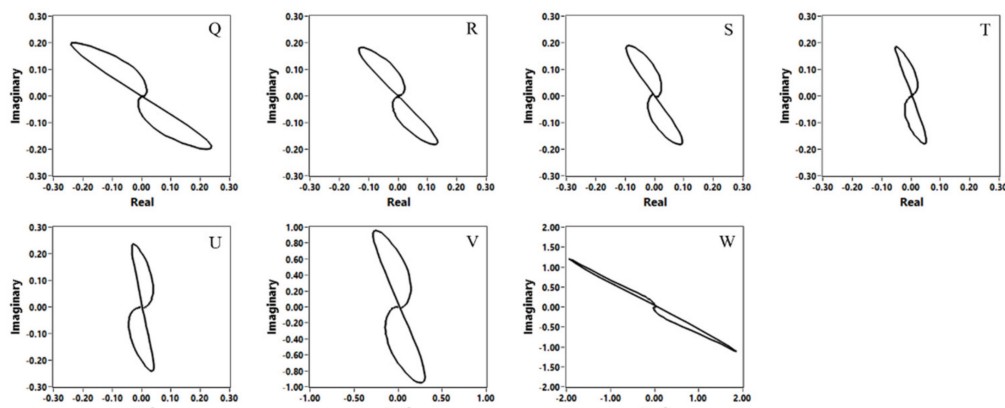

**Figure 14.** Lissajous curves from the front bobbin coils for the American Society of Mechanical Engineers (ASME) standard specimen (50 kHz AC).

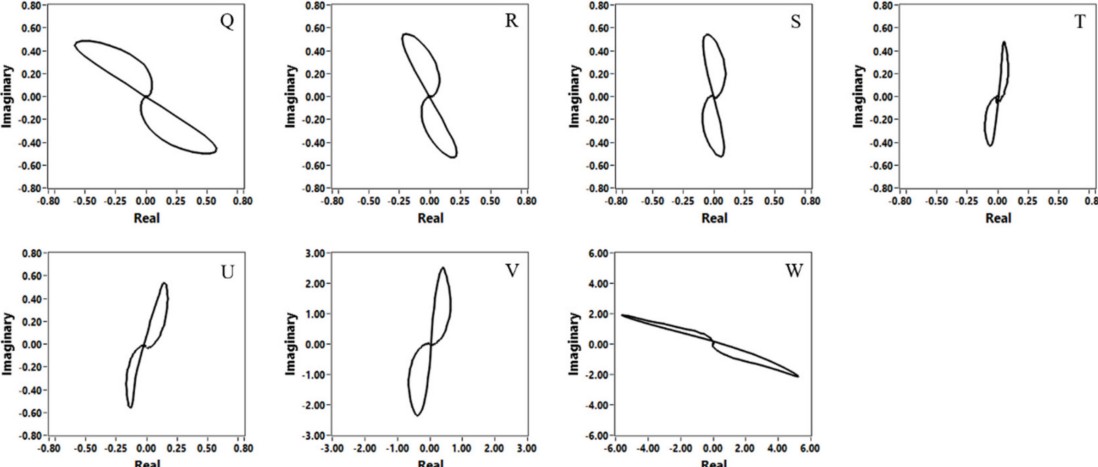

**Figure 15.** Lissajous curves from the front bobbin coils for the ASME standard specimen (100 kHz AC).

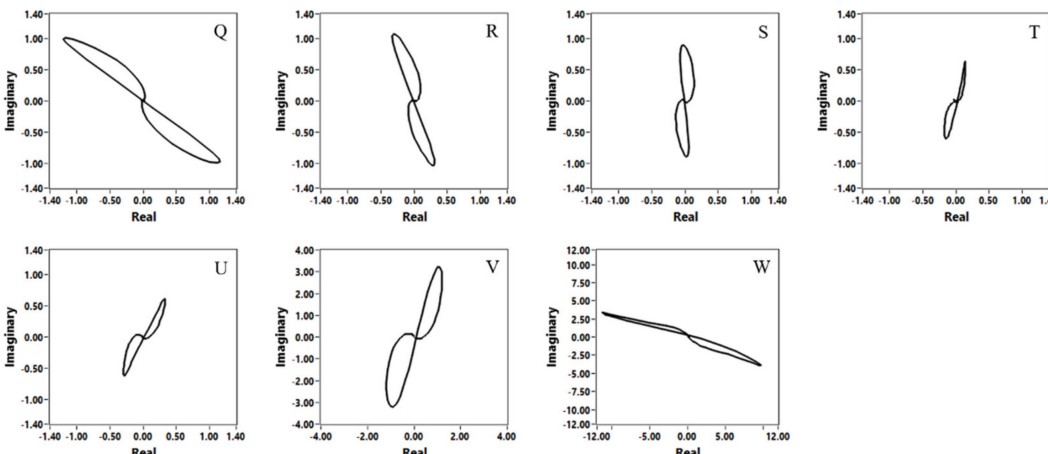

**Figure 16.** Lissajous curves from the front bobbin coils for the ASME standard specimen (150 kHz AC).

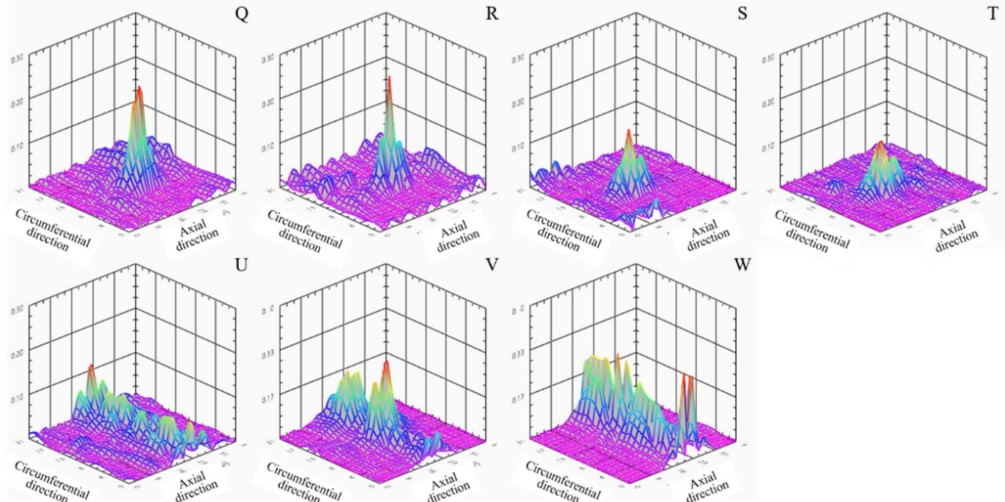

**Figure 17.** 3D image of the T-EMF from the BIGiS for the ASME standard specimen (30 kHz AC).

Figure 18a shows the phase angle changes according to the depth of the FBH. The phase angles were measured using the front bobbin coils, and 100 kHz AC was applied to the coils. As shown in Figure 18b, the depth of other flaws can evidently be evaluated quantitatively using the change of the phase angle.

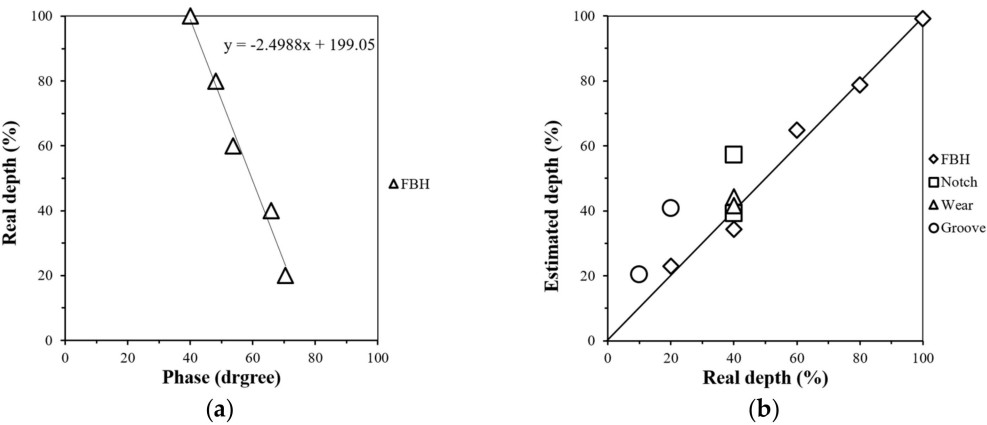

**Figure 18.** Phase angle change according to the flaw depth: (**a**) Relationship between depth and phase angle difference of the flaws (front bobbin coils, 100 kHz AC), and (**b**) quantitative analysis of depth of the flaw.

### 3.4. The Flaw Classification Algorithm

The algorithm that summarizes the test results for evaluating flaws (type, location, and depth) using the magnetic camera is as follows:

1. A dent is distinguished from other flaws by using the lobe path of the Lissajous curve obtained from the front bobbin coils.
2. From the rotational direction of the lobe with increasing frequency, the location of the flaw (ID or OD) can be determined.
3. The type of the defect is determined from the area of the lobe and the BIGiS image.
4. The multisite damage is determined from the BIGiS image.
5. The depth of the flaw is estimated using quantitative evaluation results according to location and type of defect.

The flow chart of the flaw classification algorithm is shown in Figure 19.

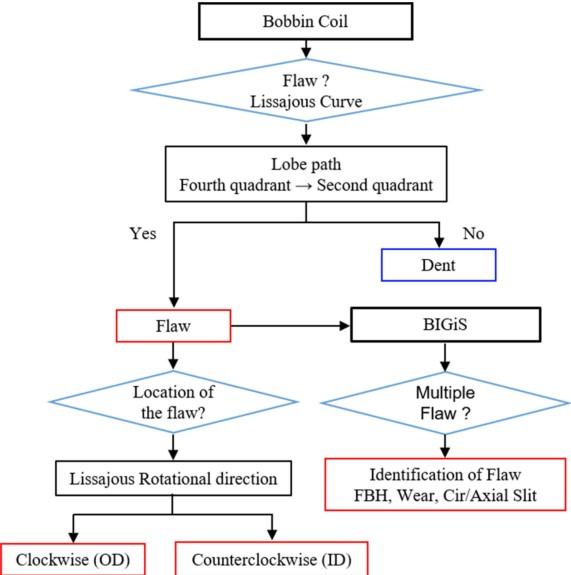

**Figure 19.** The flaw classification algorithm.

### 4. Conclusions

This study proposed an algorithm that can evaluate the presence, location, shape, and depth of the flaws in HP FWH tubes using a bobbin type magnetic camera composed of a differential-type bobbin coil and a GMR sensor array. The characteristics of the output signal of the differential-type bobbin coils and the GMR sensor array according to the presence, location, and shape of the defects were investigated using various types of artificial flaws introduced to small-bore tubes made of austenitic stainless steel. The presence of flaws can be checked using the lobe path direction of the Lissajous curve, and the location of the flaw can be determined from the rotation direction of the lobe according to the input frequency. The shape of the flaw can be determined from the area of the lobe and the T-EMF image of the GMR sensor array. Additionally, multisite damages can be determined from the T-EMF image. Notches with more than 40% OD depth or 20% ID depth can be evaluated quantitatively by using the phase angle change.

**Author Contributions:** Conceptualization, J.L.; methodology, J.L., S.S., H.L. (Hoyong Lee) and H.L. (Heejong Lee); software, S.S.; validation, J.L., S.S., and H.L. (Heejong Lee); formal analysis, S.S., and H.L. (Hoyong Lee); investigation, J.L. and S.S.; resources, J.L. and H.L. (Heejong Lee); data curation, S.S. and H.L. (Hoyong Lee); writing—original draft preparation, S.S. and H.L. (Hoyong Lee); writing—review and editing, H.L. (Heejong Lee) and J.L.; visualization, S.S.; supervision, J.L.; project administration, J.L.; funding acquisition, J.L. This paper was prepared with the contribution of all authors.

**Funding:** This research was funded by the Korea Institute of Energy Technology Evaluation and Planning, grant number KETEP 20171520101610.

**Acknowledgments:** We are grateful for the support of the Korea Institute of Energy Technology Evaluation and Planning (KETEP).

**Conflicts of Interest:** The authors declare no conflict of interest.

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
