# Peer review of "Flaw Classification Algorithm for Heat Exchanger Tubes Using a Bobbin-Type Magnetic Camera"

_applsci, doi:10.3390/app9235000_

Round 1

Reviewer 1 Report

Paper is interesting and can be published considering some points as follows:

The Figure 3 should be put after the descriptions.

Figure 5 is completely unclear.

Same Figure 7.

Same Figure 8.

What are the challenges of current study and how authors would like to deal with them as direction for future.

The shortages of current study.

Providing a framework at beginning of study may helpful.

Author Response

Thank you for taking time reviewing our paper! We have revised the paper according to your helpful comments. The changes were "Track Changed" in the revision version.

Point 1: The Figure 3 should be put after the descriptions.

Response 1: We changed the position of Figure 3 so that the figure appears after the description.

Point 2: Figure 5, 7, 8 are completely unclear.

Response 2: We changed Figure 5a-d, 7a-d, and 8a-d to Figure 5-8, Figure 10-13, and Figure 14-17, and enlarged the Figures to make them clear. Also, we changed the angle of the T-EMF plots to improve visibility.

Point 3: What are the challenges of current study and how authors would like to deal with them as direction for future. The shortages of current study. Providing a framework at beginning of study may helpful.

Response 3: Thank you for your comments! In accordance with your recommendations, the last paragraph of the introduction has been revised as follows:

“Lee et al. developed bobbin-type magnetic cameras to quickly access large number of HX tubes [2-8]. In order for magnetic camera technology to be effectively applied to industrial field, an algorithm should be provided to make it easier for inspectors to identify and evaluate flaws. This study is a follow up project of the bobbin-type magnetic camera technology development by Lee et al. It proposes an algorithm to identify the location, recognize the shape, and quantitatively evaluate the depth of flaws when inspecting HP FWH tubes of austenitic stainless steel using a bobbin-type magnetic camera. The effectiveness of the proposed algorithm was evaluated using three test specimens that represent different types of flaws, ASME test standard, and multisite damage.”

Reviewer 2 Report

This paper proposed an algorithm to evaluate the presence, location, shape, and depth of the flows in HP FWH tubes by using a differential-type bobbin coils and GMR sensor array. The experiment is well designed and the results are convincing. Thus I suggest to accept this paper after the following adjustments:

Correct Figure 5(c), 7(c), and 8(c) captions, they should be “Lissajous curves from the front bobbin coils (150 kHz AC)” Please increase the x-y axis tick label font size in Figures 5, 7, 8. They are unreadable. In Figure 6, there is a typo of “Specimen”, please correct. 

Author Response

Thank you for taking time reviewing our paper! We have revised the paper according to your helpful comments. The changes were “Track Changed” in the revision version.

Point 1: Correct Figure 5(c), 7(c), and 8(c) captions, they should be “Lissajous curves from the front bobbin coils (150 kHz AC)” Please increase the x-y axis tick label font size in Figures 5, 7, 8. They are unreadable.

Response 1 : Thank you for your correction! We corrected captions of Figure 5(c), 7(c) and 8(c) as your comment. Also, we changed Figure 5a-d, 7a-d, and 8a-d to Figure 5-8, Figure 10-13, and Figure 14-17, and enlarged the Figures to make them clear. Also, we changed the angle of the T-EMF plots to improve visibility.

Point 2: In Figure 6, there is a typo of “Specimen”, please correct.  

Response 2 : The typo in Figure 6 was corrected.

Reviewer 3 Report

In fig. 5-8 signatures for the designations (b) and (c) are the same:

"Lissajous curves from the front bobbin coils (100 kHz AC)".

Probably the designation (s) should correspond to (150 kHz AC)?

Author Response

Thank you for taking time reviewing our paper! We have revised the paper according to your helpful comments. The changes were “Track Changed” in the revision version.

Point 1: In fig. 5-8 signatures for the designations (b) and (c) are the same:

Response 1: Thank you for your correction! We corrected captions of Figure 5(c), 7(c) and 8(c) as your comment.